Chronic intake of high-dose of blueberry leaf extract does not augment the harmful effects of ethanol in rats

Yamasaki Kaede 1
Sugamoto Kazuhiro 1
Arakawa Teruaki 2
Nishiyama Kazuo 1
Yamasaki Masao myamasaki@cc.miyazaki-u.ac.jp 1
1 Interdisciplinary Graduate School of Agriculture and Engineering, University of Miyazaki , Miyazaki , Miyazaki , Japan
2 Bizen Chemical Company Limited , Akaiwa , Okayama , Japan
Uversky Vladimir
Electronic publication date: 2019 Jun 7
Publication date: 2019
Volume: 7
Electronic Location ID: e6989
Received 2019 Jan 25; Accepted 2019 Apr 20
Copyright: ©2019 Yamasaki et al.
Copyright year: 2019
Copyright holder: Yamasaki et al.
License: This is an open access article distributed under the terms of the Creative Commons Attribution License, which permits unrestricted use, distribution, reproduction and adaptation in any medium and for any purpose provided that it is properly attributed. For attribution, the original author(s), title, publication source (PeerJ) and either DOI or URL of the article must be cited.
License URL: https://creativecommons.org/licenses/by/4.0/

Keywords: Blueberry leaf extract, Alcohol metabolism, Ethanol

Funding: The authors received no funding for this work.

==============================
Excessive alcohol consumption is a risk factor for liver diseases. Enhancement of alcohol metabolism could be an effective strategy to prevent these adverse effects since it promotes the clearance of ethanol and acetaldehyde from the serum. Polyphenol-rich products have shown to protect against alcohol-related liver damage. Blueberry leaves have attracted attention as they are rich polyphenols such as proantocyanidins and chlorogenic acid. In this study, we investigated the effects of a high dose of blueberry leaf extract (BLEx) on alcohol metabolism during chronic intake of ethanol. Seven-week old Sprague-Dawley (SD) rats were divided into four groups: normal liquid diet group (NLD), normal liquid diet + BLEx group (NLD + BLEx), alcohol liquid diet group (ALD), and alcohol liquid diet + BLEx (ALD + BLEx). Then, rats were fed experimental diet for 5 weeks and at the end of feeding period, body weight, food intake, liver weight, indices of liver injury, expression and activity of alcohol metabolism-related and anti-oxidative enzymes, and levels of carbonyl protein, triglyceride (TG), and total cholesterol (T-Chol) were measured. Body weight and food intake decreased, whereas liver aldehyde dehydrogenase (ALDH) activity, liver microsomal cytochrome P450 2E1 (CYP2E1) protein and mRNA expression, and heme oxygenase 1 (HO-1) mRNA expression were upregulated by ethanol intake. Dietary BLEx, however, did not affect any of these ethanol-related changes. Indices of liver injury, expression and activity of other alcohol metabolism-related enzymes, liver carbonyl protein, TG, and T-Chol levels were not altered by ethanol and BLEx. Thus, chronic BLEx intake does not ameliorate the harmful effects of ethanol.

Introduction

Ethanol is metabolized to acetaldehyde by liver alcohol dehydrogenase (ADH) and liver microsomal cytochrome P450 2E1 (CYP2E1). It is further metabolized to acetic acid by liver mitochondrial aldehyde dehydrogenase (ALDH). Finally, acetic acid is decomposed into water and carbon dioxide by the tricarboxylic acid cycle and then excreted from the body (Zakhari, 2006). A substantial proportion of Asians, including Japanese, have mutations in the genes coding for ADH and ALDH resulting in an impaired ethanol metabolism (Eng, Luczak & Wall, 2007). Thus, they often experience unpleasant side effects such as headache and nausea after excessive alcohol intake. The World Health Organization (WHO) has reported that alcohol abuse is the third leading risk factor for liver disease worldwide (World Health Organization, WHO), and presented a “global strategy to reduce the harmful use of alcohol” in 2010 (World Health Organization, WHO). Indeed, ethanol abuse substantially increases the risk of liver disease (Setshedi, Wands & Monte, 2010), acute lung injury (Kaphalia & Calhoun, 2013), and carcinogenesis (Seitz & Stickel, 2007). Enhancement of alcohol metabolism could be an effective strategy to prevent these effects since it promotes the clearance of ethanol and acetaldehyde from the serum. The metabolism of alcohol after chronic consumption has been shown to increase with noni juice containing a high concentration of proanthocyanidin, a class of polyphenol (Chang et al., 2013). Furthermore, resveratrol, which is also a polyphenol, alleviates alcoholic fatty liver by the upregulation of sirtuin 1 and adiponectin (Ajmo et al., 2008). Heme oxygenase 1 (HO-1) is one of the antioxidant enzymes and is upregulated in response to ethanol induced oxidative stress and additional upregulation of HO-1 results in alleviation of the hepatic oxidative stress. Actually, quercetin prevents the liver from acute alcoholic injury by upregulation of HO-1 (Liu et al., 2018). Thus, polyphenol-rich products have shown the potential to protect against alcohol-related liver damage. Blueberry leaves have attracted attention as it contains novel functional components, including quinic acid, proanthocyanidins, and several polyphenols (Matsuo et al., 2010). Further, blueberry leaves have been reported to exert protective effects against fatty liver (Yuji et al., 2013). It also has anti-fibrogenic (Takami et al., 2010) and suppressive effects on hepatitis C virus replication (Takeshita et al., 2009). We have confirmed that a single dose of blueberry leaf extract (BLEx) after single-dose ethanol reduces serum ethanol level (Yamasaki et al., 2016). From this previous data, we have hypothesized that BLEx could prevent the gastro intestinal absorption. The effect of chronic intake of a high dose of BLEx is, however, not clear. In this study to verify the safety of BLEx, we investigated the effects of high-dose BLEx on alcohol metabolism during chronic intake of ethanol.

Materials & Methods

Reagents

BLEx was prepared as a hot water extract by Bizen Chemical Co. Ltd (Okayama, Japan). Briefly, blueberry leaf powder was extracted in 16 parts of hot water (95–100 ° C) for 30 min twice. Then, the extract was filtered and heat sterilized. Finally, the extract was dried with a spray dryer, producing a powder. Ethanol was purchased from the Wako Pure Chemical Industries (Osaka, Japan). The oxidized form of β-nicotinamide adenine dinucleotide hydrate (NADH), 4-methylpyrazole, 2,4-dinitrophenylhydrazine, and trifluoroacetic acid (TFA) were purchased from the Tokyo Chemical Industry (Tokyo, Japan).

Ethanol intake model

The animal studies were conducted in accordance with the Guide for the Care and Use of Laboratory Animals of the University of Miyazaki (Animal Experiment Committee of Miyazaki University: 2017-014-2) and in compliance with the Law Concerning the Protection and Control of Animals (Japan Law No. 105), Standards Relating to the Care and Management of Laboratory Animals and Relief of Pain (Notification no. 88 of the Ministry of the Environment, Japan), and The Guidelines for Animals Experimentation (the Japanese Association for Laboratory Animal Science). The room temperature was maintained at 22–24 °C and the animals were housed under a 12-h light/dark cycle (09:00–21:00).

Sprague-Dawley (SD) rats (all male, 7 weeks old, total 20) were purchased from Japan SLC (Hamamatsu, Japan) and acclimatized for 1 week. The rats were divided into 4 groups (5 per group): normal liquid diet group (NLD), normal liquid diet + BLEx group (NLD + BLEx), alcohol liquid diet group (ALD), and alcohol liquid diet + BLEx group (ALD + BLEx). The rats were fed with a Lieber-DeCarli liquid diet (Lieber & DeCarli, 1982). The composition of the liquid diet is shown in Table 1. Ethanol intake in the ALD group was gradually increased from 1% on day 1 to 5% over 7 days. 3% BLEx liquid diet was mixed with dry diet and the rats were fed for 5 weeks. After the end of the experiment, the rats were sacrificed using a mixture medetomidine (Kyoritsu Seiyaku Corporation, Tokyo, Japan), midazolam (Astellas Pharma Inc., Tokyo, Japan), and butorphanol (Meiji Seika Pharma Co., Ltd., Tokyo, Japan).

Table 1 Composition of the liquid diet used in this study.

Composition of the liquid diet used in this study (g/L liquid diet).

	NLD	NLD + BLEx	ALD	ALD + BLEx	
Casein Na	41.4	41.4	41.4	41.4	
L-cystine	0.5	0.5	0.5	0.5	
DL-methionine	0.3	0.3	0.3	0.3	
Corn oil	8.5	8.5	8.5	8.5	
Olive oil	28.4	28.4	28.4	28.4	
Safflower oil	2.7	2.7	2.7	2.7	
Vitamin Mix	2.5	2.5	2.5	2.5	
Mineral Mix	8.75	8.75	8.75	8.75	
Dextrin	115.2	108.5466	25.6	21.6346	
Cellulose	10.0	10.0	10.0	10.0	
Choline bitartrate	0.53	0.53	0.53	0.53	
Xanthan gum	3.0	3.0	3.0	3.0	
BLEx	0.0	6.6534	0.0	3.9654	
total (in dry diet)	221.78	221.78	132.18	132.18	
Ethanol	0.0	0.0	50.0	50.0	

Serum biochemical tests

Serum alanine aminotransferase (ALT) and aspartate aminotransferase (AST) levels were estimated using Wako Transaminase CII-Test Kit (Wako). Serum total protein and albumin level and albumin/globulin (A/G) ratio were estimated using A/G B-Test Wako Kit (Wako).

Liver aldehyde dehydrogenase (ADH) and alcohol dehydrogenase
(ALDH) activities

The liver tissues (50 mg) were homogenized using 250 µl of homogenized buffer (0.25 M sucrose, 5 mM Tris, 0.5 mM ethylenediaminetetraacetic acid (EDTA)-2Na, and 2 mM 2-mercaptoethanol). The homogenates were centrifuged (13, 000 × g for 10 min at 4 °C) and the supernatant was collected. Ten-fold diluted supernatant (25 µl) was mixed with ADH reaction assay buffer (225 µl, 5 mM NADH, 5 mM ethanol, 9 mM glycine, and 120 mM sodium pyrophosphate (pH 8.8)) or ADH negative assay buffer ((225 µl, 5 mM NADH, 1 mM 4-methylpyrazole, 9 mM glycine, and 120 mM sodium pyrophosphate (pH 8.8)) in a 96-well plate. To determine ALDH activity, 25 µl of 10-fold diluted supernatant was mixed with 225 µl of ALDH reaction assay buffer [0.5 mM NADH, 10 mM acetaldehyde, 0.1 mM 4-methylpyrazole, 2 µM rotenone, and 60 mM sodium pyrophosphate (pH 8.8)] or 225 µl ALDH negative assay buffer (0.5 mM NADH, 0.1 mM 4-methylpyrazole, 2 µM rotenone, and 60 mM sodium pyrophosphate (pH 8.8)) in a 96-well plate. The change in absorbance (340 nm) was measured by a plate reader at 0, 3, 6, 9, 12, and 15 min. The protein level was estimated by using a commercial Pierce® Bicinchoninic Acid (BCA) Protein Assay Kit (Thermo). The liver ADH and ALDH activities were calculated by the following equation: ADH and ALDH activity = (ΔA/min × V × D)/(6. 3 × d × v), where, ΔA/min is the change in absorbance at 340 nm, V is the final volume, D is the dilution rate, 6.3 is the molecular extinction coefficient at 1 mM of NADH at 340 nm, d is the optical path length, and v is the sample volume.

Fractionation of the liver microsomal fraction

The liver tissues (2 g) were homogenized using 10 ml of homogenized buffer (250 mM sucrose, 75 mM nicotinamide, 2.5 mM EDTA-2Na, 20 mM 2-mercaptoethanol, and 50 mM potassium phosphate buffer (pH 7.4)). Following that, the homogenates were centrifuged at 12, 425 × g for 20 min at 4 °C and the supernatants were ultracentrifuged at 109, 572 × g for 1 h at 4 °C. After the supernatant was discarded, the pellets were redissolved in 2 ml of dissolution buffer (5 mM (±)-dithiothreitol and 20 mM potassium phosphate buffer (pH 7.4)).

Western blot

The microsomal fraction was lysed with 50 mM Tris-HCl (pH 7.5) containing 150 mM NaCl, 2% Triton X-100, 2 mM EDTA, 50 mM NaF, and 30 mM Na4P2O7 with 1/50 volume of a protease inhibitor cocktail (Nacalai Tesque, Kyoto, Japan). The protein levels were estimated using a BCA protein assay reagent (Pierce, Rockford, IL). The lysate containing 1 µg of protein was denatured and separated by electrophoresis on a 10% sodium dodecyl sulfate-polyacrylamide gel and transferred onto Hybond®-P polyvinylidene fluoride (PVDF) membranes (GE Healthcare, Buckinghamshire, UK). The nonspecific sites were blocked by incubating the membrane with 3% non-fat dried milk in Tris-buffered saline and 0.1% Tween-20 (T-TBS) for 60 min at a room temperature. The antibodies were diluted with Can Get Signal solutions 1 and 2 (TOYOBO, Tokyo, Japan). CYP2E1 specific polyclonal antibody was purchased from Proteintech Group, Inc. (IL, USA). Horseradish peroxidase-conjugated anti-rabbit IgG was purchased from Cell Signaling Technology® (Cell Signaling Technology, Inc., MA, USA). Following each antibody binding reaction, the membranes were washed with T-TBS. The proteins on the membrane were detected using ImageQuant LAS 4000 (GE Healthcare). The band intensity was quantified using ImageQuant TL (GE Healthcare) and the representative blot patterns are shown.

Isolation of RNA, reverse-transcription of RNA, and real-time polymerase chain reaction (PCR)

The total RNA was isolated from the liver using TRIzol reagent (Thermo Fisher Scientific Inc., MA, USA). Rever Tra Ace qPCR RT Kit (TOYOBO) was used to synthesize cDNA. Real-time PCR was performed with an Agilent AriaMx Real-Time PCR System (Agilent Technologies, Inc., CA, USA) using THUNDERBIRD® SYBR® qPCR Mix (TOYOBO). The temperature was controlled according to the manufacturer’s instructions. The primer sequences for the real-time PCR are listed in Table 2. To quantify mRNA expression, the data obtained by real-time PCR were analyzed according to the Praffl method. Expression of β-actin, a housekeeping gene was estimated to normalize the expression of the target genes.

Table 2 Primer sequences used for the real-time PCR.

Target gene	Primer	Nucleotide sequence	
ADH1	Forward	5′-CCTTCACCATCGAGGACATA-3′	
	Reverse	5′-GCCACCATCTTAATGCGAACTT-3′	
ALDH2	Forward	5′-GTGTTCGGAGACGTCAAAGA-3′	
	Reverse	5′-GCAGAGCTTGGGACAGGTAA-3′	
CYP2E1	Forward	5′-CCTACATGGATGCTGTGGTG-3′	
	Reverse	5′-CTGGAAACTCATGGCTGTCA-3′	
HO-1	Forward	5′-TGGCCCACGCATATACCCGCT-3′	
	Reverse	5′-TTGAGCAGGAAGGCGGTCTTAG-3′	
β-actin	Forward	5′-GAGCTATGAGCTGCCTGACG-3′	
	Reverse	5′-GGATGTCAACGTCACACTTC-3′	

Estimation of liver carbonyl protein

The liver carbonyl protein assay was performed according to the method described by Colombo et al. (Colombo et al., 2016). The liver tissues (100 mg) were homogenized using 1 ml phosphate buffered saline (PBS), the homogenates were centrifuged (12,000× g for 15 min at 4 °C), and the supernatant was collected. The protein level in the supernatant was determined using a BCA protein assay reagent. The supernatant was diluted to a concentration of 1 mg/ml. 500 µl of the diluted supernatant was mixed with 100 µl of 10 mM 2,4-dinitrophenylhydrazine (DNPH)-(2N) HCl and incubated for 1 h at a room temperature with shaking.

The final sample was mixed with ice-cold 20% perchloric acid (600 µl) and then incubated for 15 min on ice. Following that, the samples were centrifuged (10,000× g for 5 min at 4 °C) and the pellets were collected. The pellets were mixed with ice-cold 20% perchloric acid (500 µl) and again centrifuged (10,000× g for 5 min at 4 °C). Subsequently, the pellets were collected and mixed with 1:1 (v/v) ethanol-ethyl acetate. They were again centrifuged (10,000× g for 5 min at 4 °C) and the final pellets were collected. This process of mixing ethanol-ethyl acetate and centrifugation was repeated.

The obtained pellets were mixed with 250 µl of 0.2% (w/v) SDS in 20 mM Tris–HCl (pH 6.8) and incubated at 95 °C for 10 min. The pellets were then homogenized by ultrasonication and the protein content was determined by BCA protein assay. Western blot was performed with the sample. The samples containing 1 µg of protein were denatured. Anti-DNPH was purchased from Thermo Fisher Scientific Inc. (Waltham, MA, USA).

Estimation of liver triglyceride (TG) and total cholesterol (T-Chol) levels

The liver tissues (200 mg) were homogenized using 1 ml of PBS, the homogenates were centrifuged (12, 000 × g for 15 min at 4 °C), and the supernatant was collected. Liver TG and T-Chol levels were determined using the Triglyceride E-Test Wako (Wako) and the Cholesterol E-Test Wako (Wako), respectively.

Statistical analysis

Initially, the data were analyzed using two-way analysis of variance (ANOVA) to identify the effects of alcohol, BLEx, and their interaction. When the interaction was significant, the data were analyzed using the Tukey-Kramer test. The analyses were conducted using Statcel3 software (OMS Publishing, Saitama, Japan). A p value of <0.05 was considered as statistically significant.

Results

Effects of BLEx on body weight, food intake, and liver weight

As shown in Fig. 1, final body weight and food intake in the alcohol intake groups were significantly lower than the non-alcohol intake groups. No significant difference was, however, noted between the non-BLEx intake and BLEx intake groups. The liver weight was unchanged by alcohol and BLEx intake.

Figure 1 Body weight, food intake and liver weight.

(A) Changes in body weight, (B) final body weight, (C) food intake change, (D) daily food intake, and (E) liver weight in rats (n = 5). The values are expressed as the mean ± SD.

Effects of BLEx on indices of alcoholic liver injury

The levels of serum AST, ALT, albumin, total protein, AST/ALT ratio, and A/G ratio in rats are shown in Fig. 2. As shown in Figs. 2A–2C, the serum ALT level in alcohol intake groups was significantly higher than the non-alcohol intake groups. The serum AST level and the ratio of AST/ALT, however, were not changed by alcohol and BLEx. As shown in Figs. 2D–2F, no significant difference was found among the experimental groups.

Figure 2 Indices of alcoholic liver injury.

(A) Serum AST level, (B) ALT level, (C) AST/ALT ratio, (D) albumin level, (E) total protein level, and (F) albumin/globulin ratio in rats (n = 5). The values are expressed as the mean ±  SD.

Effects of BLEx on level or expression of protein and mRNA related to alcohol metabolism in the liver

The liver ADH and ALDH activities in the rats are shown in Figs. 3A and 3B. The liver ADH activity was unchanged by alcohol and BLEx (Fig. 3A). The liver ALDH activity, however, significantly increased in the alcohol intake groups as compared to the non-alcohol intake groups. However, the ALDH activity was not changed by BLEx intake (Fig. 3B).

Figure 3 Level or expression of protein and mRNA related to alcohol metabolism in the liver.

(A) Liver ADH and (B) ALDH activities; liver (C) ADH1, (D) ALDH2, and (G) CYP2E1 mRNA expression; and (E and F) liver microsomal CYP2E1 protein expression in rats (n = 5). The values are expressed as the mean ± SD. ∗∗p < 0.01 compared to the NLD group, and ##p < 0.01 compared to the NLD + BLEx group.

The liver ADH1 and ALDH2 mRNA expression in the rats are shown in Figs. 3C and 3D. Liver ADH1 mRNA expression was significantly changed by BLEx intake. Liver ALDH2 mRNA expression was not changed by alcohol and BLEx. Liver CYP2E1 protein and mRNA expression levels are shown in Figs. 3E–3G. As shown in Figs. 3E and 3F, liver CYP2E1 protein expression significantly increased in the ALD and ALD + BLEx groups as compared to the NLD and NLD + BLEx groups, respectively. No significant difference in the liver CYP2E1 protein expression was, however, noted between the ALD and ALD +BLEx groups. As shown in Fig. 3G, the liver CYP2E1 mRNA expression was significantly higher in the alcohol intake groups as compared to the non-alcohol intake groups. However, the CYP2E1 mRNA expression was not changed by BLEx.

Effects of BLEx on liver carbonyl protein level and HO-1 mRNA expression in the liver

As shown in Fig. 4A, the liver carbonyl protein level was not changed by alcohol and BLEx. As shown in Fig. 4B, the HO-1 mRNA expression in the liver was significantly increased in the alcohol intake groups as compared to the non-alcohol intake groups.

Figure 4 Liver carbonyl protein level and HO-1 mRNA expression in the liver.

(A) Liver carbonyl protein level and (B) HO-1 mRNA expression in rats (n = 5). The values are expressed as the mean ± SD.

Effects of BLEx on liver TG and T-Chol levels in the liver

As shown in Figs. 5A and 5B, the TG and T-Chol levels in the liver were not changed by alcohol and BLEx.

Figure 5 Liver TG and T-Chol levels in the liver.

(A) Liver triglyceride and (B) total cholesterol level in rats (n = 5). The values are expressed as the mean ± SD.

Discussion

Our previous study has investigated the effects of single-dose BLEx (1 g/kg body weight) on alcohol metabolism and absorption of ethanol in single-dose ethanol intake rat (Yamasaki et al., 2016). We have previously done study in which 1 g BLEx significantly prevented elevation of serum ethanol and acetaldehyde levels after 0.5 g/kg body weight ethanol intake (M Yamasaki, 2015, unpublished data). In this study, we have investigated the effects of high-dose BLEx on alcohol metabolism in chronic ethanol intake rats. Our concerns in this study is the synergistic adverse effects of ethanol and BLEx as is some natural plant extracts regulated the expression of CYPs. For instance, extract of Ginkgo biloba upregulated CYP2E1 expression (Sugiyama et al., 2004). Information for the synergistic effect on detoxication metabolism may be an important for the prevention of the adverse effects. Based on our previous single dose studies, we set the dose of 3% BLEx. The animal model was created according to the methods described by Reyes-Gordillo et al. (Reyes-Gordillo et al., 2016). The serum AST, ALT, total protein, albumin levels, and A/G ratio, which are the indices of liver injury did not change when the rats were fed with 5% ethanol for 5 weeks, demonstrating that there was no ethanol-induced liver injury. This further signified that the amount and duration of ethanol intake were not enough to induce liver injury. A previous study has shown that alcohol intake did not affect liver weight (Yun et al., 2007), but it reduced the body weight (Rouach et al., 2005). The results of our study are similar to those in this study.

Although BLEx prevented the gain in body weight, the alcohol-induced reduction in body weight was not affected by BLEx. This indicated that BLEx did not affect alcohol-induced body weight reduction. A previous study has shown that the food intake in 4 week-old SD rats was reduced by 8 weeks intake of ethanol (Kim et al., 2014). In this study, BLEx did not affect the alcohol-induced reduction in food intake. Taken together, we have shown that a long-term BLEx intake does not affect alcohol-induced change in the food intake and body composition.

In this study, we have also investigated the effects of BLEx on ADH, ALDH, and CYP2E1, which are responsible for alcohol metabolism in the liver (Zakhari, 2006). There was a greater change in the CYP2E1 expression than the liver ADH and ALDH activities. The results are similar to a previous study which showed that CYP2E1 expression was increased by long-term alcohol intake (Cederbaum, 2010). CYP2E1 protein and mRNA expression were, however, not changed by BLEx in this study. The polyphenol level of BLEx was 403 mg/g equivalent tannic acid, whereas, the polyphenol level of the liquid diet was 1.598 g/L.

The effects of dietary polyphenol on CYP2E1 regulation has been reported. The administration of red wine containing 350 mg polyphenol/L for 9 weeks to SD rats increased CYP2E1 protein expression (Cowpland et al., 2006). On the contrary, administration of red wine containing 55.2 mg total flavonols/L to Wister rats for 10 weeks inhibited alcohol-induced CYP2E1 protein expression (Orellana et al., 2002). It was also demonstrated that an intake of 3 g/L of dietary epigallocatechin-3-gallate did not affect the alcohol-induced increase in CYP2E1 protein expression (Yun et al., 2007). The human equivalent dose of BLEx used in this study (1.69 g/kg) (U.S. Food and Drug Administration (FDA), 2005) was very high. In addition, the polyphenol content in the liquid diet (1.60 g/L) was also much higher than used in other studies. BLEx, however, did not affect alcohol-induced increased CYP2E1 protein and mRNA expression in the liver.

Proanthocyanidins (11.34%) is present in the blueberry leaves (Matsuo et al., 2010). When experimental rats were administered 12.6 mg/kg proanthocyanidins for 5 days, the liver CYP2E1 activity was found to be unchanged (Sugiyama et al., 2004). This study shows that chronic intake of a high dose BLEx did not affect liver CYP2E1 expression. In this study, liver ADH activity and ADH1 and ALDH2 mRNA expression were not changed by alcohol intake, although the liver ALDH activity was significantly increased. Similar findings were previously reported by other authors (Kishimoto et al., 1995). The liver ADH and ALDH activities and ADH1 and ALDH2 mRNA expression, however, were unchanged by BLEx. A previous study has shown alcohol/dextrose meal intake decreases blood ethanol due to impairment of gastric emptying (Kaufman & Kaye, 1979). In addition, grape-seed proanthocyanidin delays gastric emptying in rat (Serrano et al., 2016). Our previous study shown that BLEx could be useful for preventing alcohol-related disorders by inhibiting ethanol absorption (Yamasaki et al., 2016). Thus, BLEx might modulate alcohol absorption through delaying gastric emptying. Therefore, long-term intake of a high dose of BLEx does not affect alcohol metabolism in the liver.

Alcohol-related oxidative stress in the liver is caused by CYP2E1 (Koop, 2006), while alcoholic liver steatosis is caused by oxidative stress (Yang et al., 2012). Because of the oxidative stress, HO-1 mRNA expression and activity are increased by augmenting CYP2E1 expression. In this study, we have found that the HO-1 mRNA and CYP2E1 protein and mRNA expression in the liver were upregulated by chronic alcohol intake.

The carbonyl protein level in the liver also increases due to alcohol-related oxidative stress (Galligan et al., 2012; Jayaraman, Veerappan & Namasivayam, 2009). In this study, we have found that liver carbonyl protein level was unchanged by alcohol, signifying that alcohol did not result in an oxidative stress because of sufficient anti-oxidant reserve. Several reports have shown that anti-oxidative protein expression can be induced by polyphenols. Dietary tea polyphenols were found to improve alcohol-induced rise in serum malondialdehyde level and decrease in serum superoxide dismutase level in rats with ethanol-induced liver fibrosis (Li et al., 2004). In addition, HO-1 protein and gene expression was found to be significantly increased by chlorogenic acid in a dose-dependent manner (Shi et al., 2018). Since dietary BLEx did not change liver HO-1 mRNA expression in this study, it can be concluded that a high dose of BLEx does not affect anti-oxidant reserve in spite of CYP2E1 induction by ethanol.

Previous studies have shown that alcoholic liver steatosis could develop due to increased CYP2E1 expression (Lu et al., 2008; Lu et al., 2010; Ceni, Mello & Galli, 2014). In this study, although liver CYP2E1 protein and mRNA levels were increased by alcohol, liver TG and T-Chol levels were not changed. Therefore, alcoholic liver steatosis was not induced by ethanol. In addition, since BLEx feeding did not change the liver TG and T-Chol levels, we inferred that a high dose of BLEx does not affect the liver lipid store.

Conclusions

Chronic alcohol intake increased the liver CYP2E1 and anti-oxidative protein expression without inducing alcohol-related oxidative stress and alcoholic liver steatosis. A chronic high-dose of BLEx did not augment the harmful effects of ethanol.

Supplemental Information

Dataset S1 Changes in body weight and final body weight dataset

Click here for additional data file.

Dataset S2 Food intake change and daily food intake dataset

Click here for additional data file.

Dataset S3 Liver weight dataset

Click here for additional data file.

Dataset S4 Serum AST and ALT level dataset

Click here for additional data file.

Dataset S5 Serum albumin and total protein level dataset

Click here for additional data file.

Dataset S6 Liver ADH activity dataset

Click here for additional data file.

Dataset S7 Liver ALDH activity dataset

Click here for additional data file.

Dataset S8 Liver ADH1 mRNA level dataset

Click here for additional data file.

Dataset S9 Liver ALDH2 mRNA level dataset

Click here for additional data file.

Dataset S10 Photo of western blot (liver microsomal CYP2E1)

Click here for additional data file.

Dataset S11 Quantified liver microsomal CYP2E1 protein dataset

Click here for additional data file.

Dataset S12 Liver CYP2E1 mRNA level dataset

Click here for additional data file.

Dataset S13 Quantified liver carbonyl protein dataset

Click here for additional data file.

Dataset S14 Liver HO-1 mRNA level dataset

Click here for additional data file.

Dataset S15 Liver TG level dataset

Click here for additional data file.

Dataset S16 Liver T-Chol level dataset

Click here for additional data file.

Additional Information and Declarations

Competing Interests

Author Contributions

Animal Ethics

Data Availability

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

The following information was supplied regarding data availability:

The raw measurements are available in Dataset S1–S16.