# Peer review of "Chronic intake of high-dose of blueberry leaf extract does not augment the harmful effects of ethanol in rats"

_PeerJ, doi:10.7717/peerj.6989_

## Round 0.1 · original submission · Major Revisions

Both reviewers have made valid comments and suggestions which I hope you will find helpful in improving your manuscript. Please write a response to each comment clearly indicating what changes you have made or justifying why you do not believe a change is required. Please pay particular attention to describing your statistical analysis. I looking forward to receiving your updated manuscript.

Reviewer 1 ·

Basic reporting

no comment

Experimental design

no comment

Validity of the findings

no comment

Additional comments

In this manuscript, effect of high-dose BLEx (blueberry leaf extract) has been investigated on alcohol metabolism during chronic intake of ethanol. Then, authors have concluded that chronic alcohol intake could increase the liver CYP2E1 and anti-oxidative protein expression, while the chronic high-dose BLEx did not augment (= modulate any) harmful effects of ethanol. It is an important area for investigating whether functional foods (or so-called health foods) can improve the disease risks and also attenuate the harmful effect of disease settings. Because the functional foods sometimes interfere with the medication, it may be necessary to demonstrate their non-toxic or no side effect in individual settings. Before publication, please consider the issues raised as following.

1. As mentioned in Introduction in lines 77-78 with authors’ previous report (Yamasaki et al., 2016), this reviewer could understand why authors wanted to investigate the effect of high-dose BLEx on alcohol metabolism during chronic intake of ethanol, although the BLEx did not show any modulating effects in this paper. To avoid concluding that results obtained here indicate just no effects, authors can emphasize the experimental differences in detail between previous work and this study in the Introduction and Discussion sections).

2. Line 85 in the Materials and Methods section: please describe how to prepare the hot aqueous extract of BLEx in detail.

3. Please describe the reason concerning the concentrations/amounts of BLEx and ethanol used in this study in text. If you have any available data of pilot study(ies) or previous paper (as mentioned above), please discuss this point.

4. Please mention about HO-1 in the Introduction section, like as stated in the Discussion section.

5. In the Figures 1A and 1C, please annotate the experimental difference of ethanol 1% and 5%, for example, with additional lines and arrows, as stated in line 103.

Others
Line 42, in the Abstract section. Typo? ‘dody’
Line 138, in the Materials and Methods section. Typo? ‘1,09,572’

Reviewer 2 ·

Basic reporting

With respect to basic reporting, the manuscript by Yamasaki et al. is well written in professional English where appropriate. In addition, there are sufficient literature references used within the context of the manuscript. Moreover, the manuscript is structured in a traditional way such that it is easy to follow. The manuscript is also self-contained such that only results relevant to the hypothesis are presented. However, more detailed are needed within the statistical section such that readers can fully understand the authors’ choice in statistical methodology. Lastly, the authors should further develop a conceptual framework explaining why they think blueberry leave extract did not aid or reduce the harmful effects of ethanol intake. Could it be a time issue where the extract might have been given first, followed by ethanol? Could the null findings be related to a species phenomenon less relevant to humans? Might it have been the chronic dosing of ethanol vs an acute effect? These concepts should be addressed in a theoretical fashion within the discussion section.

Experimental design

With respect to the experimental design used by the authors, the manuscript represents original research and methodological approaches. The research question is well defined, and presented within a sociological context of drug use. Indeed, a relevant knowledge gap is attempted to be addressed with detailed research approaches. This reviewer believes that the authors of the manuscript attempted a rigorous investigative approach to answer their research question. In addition, the research provided by Yamasaki et al. was conducted within the conformity of ethical standards within the animal research field. While methods are described in detail, the use of table 1 should be moved to the methods section for further clarification. Additionally, the primer sequences table should be moved to a supplementary section.

Validity of the findings

With respect to the research findings, Yamasaki et al. present null data where the impact of blueberry leave extract on the deleterious bio-chemical effects of ethanol is found to be inconsequential. Despite not finding significant differences within the research data, this reviewer believes it is important to disclose non-significant findings. Indeed, the merits of the work is validated by clear data provided where across multiple assays, the research found similar results. Ethanol exposure does not differ when compared to ethanol plus blueberry leave extract. However, given the low number of animals per experimental group, this reviewer suggest the authors add more rats to each group in order to reach satisfactory conclusions. Regarding the statistical approach, much work is needed. Specifically, the authors should revisit the statistical section of the manuscript and fully describe their approach. More detail is required within this section.

Additional comments

The manuscript presented by Yamasaki et al. is well written and presented. It is interesting to find that blueberry leave extract does not protect against the effects of ethanol. In addition, blueberry leave extract does not further complicate the harmful effects of ethanol in rodents. The authors also conduct an array of assays examining food consumption, protein liver enzymatic levels, and mRNA expression of liver enzymatic levels. Comments that follow are intended to improve the manuscript. Within the abstract, on line 33, the word ethanol would be changed to alcohol. On line 36 the wording should be changed to “leaves have attracted attention as they are rich”. On line, the last sentence of the abstract should be written as “Thus, chronic BLEx intake does not ameliorate the harmful effects of ethanol”.

Within the Introduction section, on line 61, the test “excessive ethanol consumption is a risk factor for liver disease (Gao et al) should be deleted. Within the methods section, on line 98, please state both the total number of animals, and rats per treatment group. On line 198, more distraction is required the statistics used. Did the researchers us an ANOVA? The results section requires much improvement. Clear statistical values are required per each assay conducted. Indeed, the text in the results section reads more like a figure legend. Furthermore, the western blot appearing in Figure 4 (A) should be redone or removed from the manuscript as the qualitative validity of the blot is poor. Lastly, similar to the paragraph on 302, the authors should continue to expand their conceptual notion explaining why blueberry leave extract did not impact the effects of ethanol at the behavioral, protein, nor mRNA level.

---

## Round 0.2 · accepted · Accept

All critiques were adequately addressed and the manuscript was revised accordingly.

# Reviewer 1 ·

Basic reporting

no comment

Experimental design

no comment

Validity of the findings

no comment

Additional comments

Issues raised by this reviewer in the previous manuscript have been accordingly improved in the current version.